# A Deep-Neural-Network-Based Decoding Scheme in Wireless Communication Systems †

**Yanchao Lei** [1], **Meilin He** [1,*], **Huina Song** [2], **Xuyang Teng** [1], **Zhirui Hu** [1], **Peng Pan** [1] **and Haiquan Wang** [1]

1   School of Communication Engineering, Hangzhou Dianzi University, Hangzhou 310018, China;
    221080021@hdu.edu.cn (Y.L.); tengxuyang@hdu.edu.cn (X.T.); huzhirui@hdu.edu.cn (Z.H.);
    panpeng@hdu.edu.cn (P.P.); tx_wang@hdu.edu.cn (H.W.)
2   Space Information Research Institute, Hangzhou Dianzi University, Hangzhou 310018, China;
    huinasong@hdu.edu.cn
*   Correspondence: meilinhe@hdu.edu.cn
†   This paper is an extended version of our paper "A DNN-based Decoding Scheme for Communication
    Transmission System over AWGN Channel". In Proceedings of the 2022 International Symposium on Wireless
    Communication Systems (ISWCS), Hangzhou, China, 19–22 October 2022.

**Abstract:** With the flourishing development of wireless communication, further challenges will be introduced by the future demands of emerging applications. However, in the face of more complex communication scenarios, favorable decoding results may not be yielded by conventional channel decoding schemes based on mathematical models. The remarkable contributions of deep neural networks (DNNs) in various fields have garnered widespread recognition, which has ignited our enthusiasm for their application in wireless communication systems. Therefore, a reliable DNN-based decoding scheme designed for wireless communication systems is proposed. This scheme comprises efficient local decoding using linear and nonlinear operations. To be specific, linear operations are carried out on the edges connecting neurons, while nonlinear operations are performed on each neuron. After forward propagation through the DNN, the loss value is estimated based on the output, and backward propagation is employed to update the weights and biases. This process is performed iteratively until a near-optimal message sequence is recovered. Various factors within the DNN are considered in the simulation and the potential impacts of each factor are analyzed. Simulation results indicate that our proposed DNN-based decoding scheme is superior to the conventional hard decision.

**Keywords:** DNN-based decoding; local decoding; activation function; loss function

## 1. Introduction

With the rapid development of technology, information exchange in society has become increasingly frequent and convenient. Digital communication, as a fundamental infrastructure for modern information exchange, has also achieved unprecedented development [1]. In today's digital communication, including mobile communication, the internet, and the internet of things, it is necessary to ensure efficient and reliable transmission of information [2]. However, in digital communication, factors such as noise and channel interference can cause errors during the transmission process. To ensure the accuracy of information during transmission, channel decoding is required for detecting and correcting errors at the receiver [3]. The importance of channel decoding is self-evident. Although traditional channel decoding schemes can effectively correct errors in the channel and have achieved some success in practical applications, there are still limitations and inefficiencies, particularly when communication scenarios become excessively complex. Implementation of mathematically based communication block models can become imprecise, thereby limiting the performance of traditional schemes. To address the aforementioned challenges,

there is growing interest in exploring new paradigms for channel decoding that differ from traditional approaches.

Deep neural networks (DNNs) have become one of the research hotspots in the field of deep learning (DL) in recent years [4]. Due to its excellent performance, DNN has been widely applied in many domains, including but not limited to image classification [5], object detection [6], speech recognition [7], and so on. The success of DNN in these areas has also sparked our interest in applying DNN to wireless communications. Compared to conventional communication system models, which require precise mathematical modeling, DNN can automatically learn the whole process from input data to output results through end-to-end learning. Therefore, manual design and adjustment of various model components is unnecessary, as DNN can learn all necessary features by itself. This makes it possible for DNN to be applied to communication systems without being limited by specific mathematical models. Moreover, DNN computations can be easily parallelized, which means they can take advantage of modern hardware accelerators such as GPUs to achieve faster training and inference speeds [8].

Inspired by the advantages of DNNs, there have been various applications in the field of wireless communications. In [9], several novel communication frameworks based on DL were investigated and their superior performance was demonstrated. In [10], the DL-based channel estimation model was analyzed and shown to not be limited to a specific signal model. It was also shown to perform close to the minimum mean square error estimation in various cases without requiring prior knowledge of channel statistics. In [11], the authors proposed a method for achieving high localization accuracy and real-time operation in vehicular networks, assisted by DNN. In [12], the DL-based physical layer communication achievements were summarized, and the capabilities of DL-based communication systems with block architecture in terms of signal compression and signal detection were demonstrated. Work [13] proposed three DNN models to address the issue of nonlinear distortions caused by power amplifiers in multiple input multiple output (MIMO) systems. A DL approach using DNNs for joint MIMO detection and channel decoding was proposed in [14], with improved performance demonstrated compared to conventional model-based receivers. In [15], a DL-based decoding scheme was presented for sparse code multiple access communication system. However, in contrast to these previous works, we provide guidance on how to apply DNN to the channel decoding block of a wireless communication system as well as for selecting appropriate DNN parameters. Favorable results are achieved by adjusting different parameters in the DNN. Specifically, in the field of channel decoding, DNN can establish a statistical model by learning a large number of data samples and then perform channel decoding operations based on this model. Compared to traditional decoding approaches, DNN can complete decoding operations more accurately.

In this study, we propose a DNN-based decoding scheme over an additive white Gaussian noise (AWGN) channel. Specifically, we first partition the signal received from the AWGN channel into multiple equally sized groups, with each group undergoing an independent local decoding process. The DNN-based decoding scheme carries out local decoding in the input layer, hidden layers, and output layer, respectively. The local decoding involves two types of operations: linear operations and nonlinear operations. A linear operation involves calculating the weighted sum of the previous layer's neuron outputs multiplied by the corresponding weights and adding the biases. A nonlinear operation involves applying a nonlinear activation function to the output of neurons. Following multiple rounds of local decoding, the loss value is calculated using the mean squared error (MSE) loss function. Secondly, employing the gradient descent (GD) optimization algorithm, the weights and biases are updated for the subsequent iteration based on the loss value. After a large number of iterations, the most reliable message sequence is recovered. Simulation results demonstrate that our proposed DNN-based decoding scheme is superior to the conventional hard decision.

The remainder of this paper is organized as follows: In Section 2, the concepts of the Hamming hard decision and the DNN model are reviewed. In Section 3, we propose a DNN-based decoding scheme, and present the principles and examples of the proposed scheme. Meanwhile, an analysis of the memory cells of the proposed scheme is conducted. Simulation results considering different activation functions, loss functions, and learning rates are presented in Section 4. In this section, we also analyze the possible reasons for the good performance of our proposed scheme and evaluate its memory cell requirements. Overall conclusions are presented in Section 5.

## 2. Preliminary

In this section, we introduce some basic knowledge about the Hamming hard decision and DNN.

### 2.1. Hamming Hard Decision

For the Hamming hard decision, the decoding principle is to compare the values of Hamming distances [16]. In general, the Hamming distance between two vectors $\boldsymbol{a}$ and $\boldsymbol{b}$ is denoted as

$$d_H(\boldsymbol{a}, \boldsymbol{b}) = \sum_{i=1}^{n} (a_i \oplus b_i) \tag{1}$$

where $\boldsymbol{a} = (a_1, a_2, \cdots, a_n)$, $\boldsymbol{b} = (b_1, b_2, \cdots, b_n)$, and $\oplus$ represents the XOR operator. For the two bits $a_i$ and $b_i$ in vectors $\boldsymbol{a}$ and $\boldsymbol{b}$, $a_i \in \{0,1\}$, $b_i \in \{0,1\}$, $1 \leq i \leq n$, the Hamming distance is equal to 0 if $a_i$ is equal to $b_i$ and 1 otherwise.

### 2.2. DNN Basics

As the number of layers in a neural network increases, the model's data processing capabilities improve. An artificial neural network is composed of multiple neurons connected to each other by edges. Neural networks consist of an input layer, multiple hidden layers, and an output layer to enable more accurate and complex pattern recognition and classification tasks. It is because the neural network contains multiple hidden layers that it is named DNN. In the DNN model, each neuron and the edges connected to it have their own specific roles. There is an activation function and a bias parameter on each neuron, and a weight parameter on the edge connected to the neuron. The activation function is a nonlinear transformation of the neuron's output [17]. The input signal is multiplied by the weight, added to the bias, and then activated to obtain the output signal. In DNN, weights and biases are used to control the strength of connections between neurons and the offset of neurons. The activation function, weights, and biases in the DNN interact with each other and, by adjusting them, the strength of connections, offsets, and output results between neurons can be controlled, thereby achieving learning and prediction of the neural network.

The structure of a typical DNN model is presented in Figure 1. Any neurons in the $(l-1)$th layer of the neural network must be interconnected with the neurons in the $l$th layer. For each layer of neurons, there exists a mapping: $\boldsymbol{x_l} = h_l(\boldsymbol{x_{l-1}}; \theta_l)$, $1 \leq l \leq M$. The mapping of each layer depends not only on the output $\boldsymbol{x_{l-1}}$ of the previous layer but also on the selection of the parameter $\theta_l$. We use $\boldsymbol{\theta} = \{\theta_1, \cdots, \theta_M\}$ to denote the selection of parameters in each layer of the neural network. The $l$th layer mapping structure has the following form:

$$h_l(\boldsymbol{x_{l-1}}; \theta_l) = f(\boldsymbol{W_l} \boldsymbol{x_{l-1}} + \boldsymbol{b_l}) \tag{2}$$

where $\boldsymbol{W_l}$ is the weight parameters and $\boldsymbol{b_l}$ is the bias parameters, and $f(\cdot)$ is an activation function.

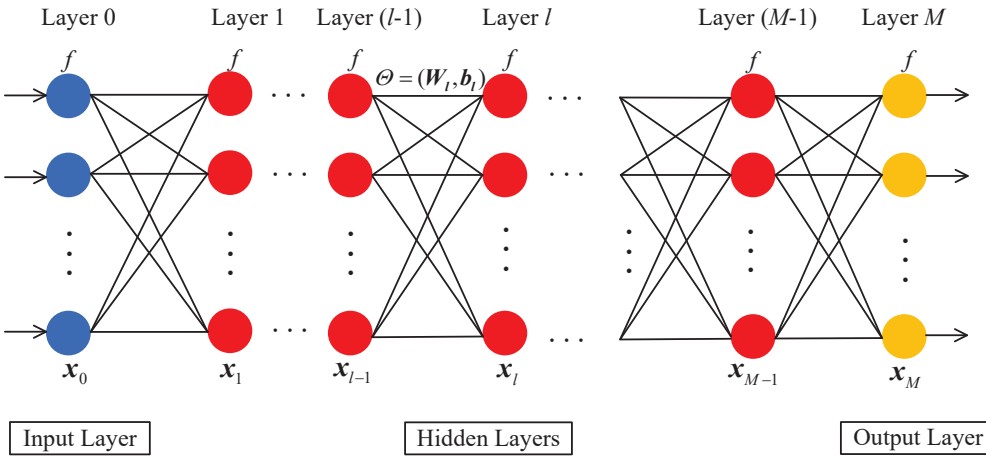

**Figure 1.** Deep neural network (DNN) structure diagram.

The forward propagation algorithm is a fundamental algorithm in DNN, responsible for transmitting input signals from the input layer to the output layer and achieving model inference and prediction. This algorithm starts from the input layer, performs layer-by-layer computation, and continues until the output layer produces the final result. The forward propagation algorithm is primarily divided into two steps: the first step involves computing the weighted sum of inputs for each neuron and the second step involves activating the input values of the neurons. For the first step, the input signal $x_j$ is multiplied by the weight $w_{ij}$ of each neuron $i$ through a weighted sum and then added with the bias $b_i$, resulting in the neuron's weighted sum expressed as Formula (3). For the second step, the activation function is applied to the weighted sum $z_i$ of each neuron $i$, resulting in the neuron's output expressed as Formula (4).

$$z_i = \sum_{j=1}^{n} w_{ij}x_j + b_i \tag{3}$$

$$a_i = f(z_i) \tag{4}$$

The following Table 1 lists some common activation functions along with their corresponding expressions and output ranges. Effective selection of the activation function is crucial for accurate predictions in DNN, as different activation functions have different usage scenarios.

**Table 1.** List of activation functions.

| Name | $f(z)$ | Range |
|:---:|:---:|:---:|
| linear | $z$ | $(-\infty, +\infty)$ |
| tanh | $\frac{e^z - e^{-z}}{e^z + e^{-z}}$ | $(-1, +1)$ |
| sigmoid | $\frac{1}{1+e^{-z}}$ | $(0, 1)$ |
| relu | $max(0, z)$ | $[0, +\infty)$ |

Compared to the forward propagation algorithm, the backward propagation algorithm is used to update the weights and biases. We employ the notation $\Theta = (W_l, b_l)$ to briefly represent the weight and bias parameters of the $l$th layer. Its basic idea is to compute the loss value through the loss function [18] and propagate the loss value layer-by-layer backwards to calculate the gradients of each neuron. Finally, the optimization function updates the $\Theta$ of each neuron based on the gradients and learning rate to gradually reduce the error [19]. The backward propagation algorithm and the forward propagation algorithm often require a significant number of iterations to achieve satisfactory results. In Table 2,

two frequently used loss functions and their corresponding expressions are listed. Note that the selection of the loss function, like that of the activation function, must be adjusted flexibly based on the training data.

**Table 2.** List of loss functions.

| Name | $E(\hat{y}_i, y_i)$ |
| --- | --- |
| MSE | $\frac{1}{n} \sum\limits_{i=1}^{n} (\hat{y}_i - y_i)^2$ |
| MAE | $\frac{1}{n} \sum\limits_{i=1}^{n} |\hat{y}_i - y_i|$ |

## 3. The Proposed DNN-Based Decoding Scheme and Memory Cell Analysis

In this section, we present our DNN-based decoding scheme and the memory cell analysis.

### 3.1. The Proposed DNN-Based Decoding Scheme

A block diagram of a communication transmission system is depicted in Figure 2. The source randomly generates a message sequence of length $K$, denoted as $t = (t_1, \cdots, t_K)$, $t_i \in \{0, 1\}, 1 \leq i \leq K$.

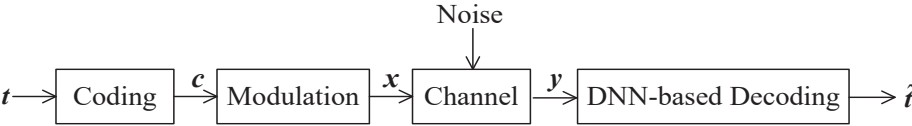

**Figure 2.** System model.

The message sequence $t$ is multiplied by the generation matrix $\mathbf{G}_{K \times N}$ to obtain the encoded vector $c = (c_1, \cdots, c_N)$. We will favor $c_i \in \{+1, -1\}$ over $c_i \in \{0, 1\}$ under the mapping $\{0 \leftrightarrow 1, 1 \leftrightarrow -1\}$, which is called binary phase shift keying (BPSK). Then, the modulated signal $x = (x_1, \cdots, x_N)$ is fed into the AWGN channel for transmission. The receiver gets a superimposed signal vector $y = (y_1, \cdots, y_N)$ with

$$y_i = x_i + z_i, 1 \leq i \leq N, \tag{5}$$

where $z_i$ is a zero-mean Gaussian variable with a variance of $\sigma^2$, i.e., $z_i \in \mathcal{N}(0, \sigma^2)$. The DNN-based decoding scheme is performed to recover the message sequence $t$. The transmission rate is $R = K/N$. The symbols and codeword synchronization are assumed.

Our DNN-based decoding scheme can also be described by the graph in Figure 3. The graph has three types of layers: the input layer in blue circles, where the input correspond to the received signal $y$; the hidden layer in red circles, where the input and output correspond to linear and nonlinear operations, respectively; and the output layer in red circles, where the output correspond to the recovered message sequence $\hat{t}$. The edge from the $i$th neuron in the $(l-1)$th layer to the $j$th neuron in the $l$th layer represents the weight parameter $W_{ij}^{l-1}$. Moreover, the bias parameter of the $j$th neuron in the $l$th layer is $b_j^l$.

The DNN-based decoding scheme is accomplished by efficient local decoding at all the neurons and interactions. During a decoding iteration, each neuron acts once to perform local decoding and updates the message on the edges to each of its adjacent neurons. A decoding iteration starts from the local decoding at the neurons in the input layer. Based on the received signal $y$, each neuron performs a local decoding. This local decoding employs a nonlinear operation by an activation function. Based on these activation function results, a local decoding is performed on each edge, which connects the neurons of the input layer and the first hidden layer. This local decoding employs a linear operation of the weights and biases. A similar process is performed in hidden layers and output layer.

Here, the sigmoid function is employed as the activation function in the hidden and output layers. In the output layer, the loss values is calculated by the MSE. Based on these loss values, the weights and biases on each edge are updated for the next iteration using the GD algorithm. After a large number of iterations, the message sequence $\hat{t}$ is recovered. It should be emphasized that the DNN-based decoding scheme provides near-optimal performance, by carrying out the local decoding iteratively, compared with hard decision.

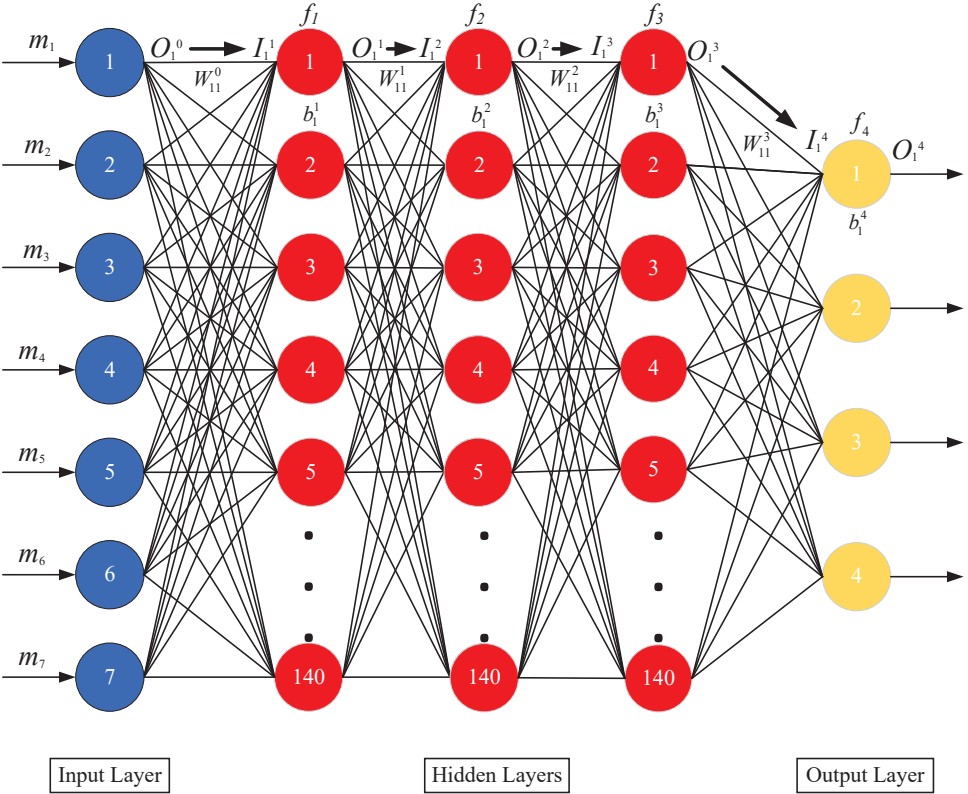

**Figure 3.** DNN-based Hamming decoding process graph.

For easy understanding, we give an example for the (7, 4) Hamming code. Based on the structure of the (7, 4) Hamming code, the numbers of neurons in the input layer and the output layer are set to 7 and 4, respectively. The signal $y$ is equally divided into multiple groups. Each group contains seven symbols, which are denoted as $m = (m_1, \cdots, m_7)$. Let

$$O_n^0 \triangleq m_n, \ 1 \leq n \leq 7, \tag{6}$$

be the output of the $n$th neuron in the input layer.

In general, given the outputs $O_n^0$ on the right side of the input layer, the overall inputs $I_j^1$ of the $j$th neuron on the left side in the first hidden layer can be represented as a function of weights and biases

$$I_j^1 = \sum_{n=1}^{7} W_{nj}^0 O_n^0 + b_j^1, \ 1 \leq j \leq 140. \tag{7}$$

Obviously, this is a linear operation.

In the first hidden layer, for a given overall input $I_j^1$, the $j$th neuron performs a nonlinear operation and, thus, its output $O_j^1$ is

$$O_j^1 = f(I_j^1) = \frac{1}{1 + e^{-I_j^1}}, \tag{8}$$

where $f(\cdot)$ is the sigmoid function.

Similarly, for the outputs $O_j^{l-1}$ of the $(l-1)$th $(2 \leq l \leq 3)$ layer, a linear operation is performed to produce the overall inputs $I_j^l$ of the $j$th neuron in the $l$th layer

$$I_j^l = \sum_{n=1}^{140} W_{nj}^{l-1} O_j^{l-1} + b_j^l. \tag{9}$$

Based on the overall inputs $I_j^l$ of the $j$th neuron in the $l$th layer, a nonlinear operation is performed to get the corresponding output,

$$O_j^l = \frac{1}{1 + e^{-I_j^l}}. \tag{10}$$

In the output layer, the same linear and nonlinear operations are performed, and thus we obtain the output $O_k^4$ of the $k$th $(1 \leq k \leq 4)$ neurons.

For the output $O_k^4$ on the right side of the output layer, the loss value $E$ is estimated by the MSE function [20]

$$E = \frac{1}{4} \sum_{k=1}^{4} (O_k^4 - t_k)^2, \tag{11}$$

where $t_k$ is a length-4 sequence of a group equally divided by $t$, based on the (7, 4) Hamming code.

Based on this estimation and the GD algorithm [21], the update rules of weights and biases on each edge are

$$\begin{cases} W_{nj}^{l-1} = W_{nj}^{l-1} - \eta \dfrac{\partial E}{\partial W_{nj}^{l-1}} & (12) \\[3mm] b_j^l = b_j^l - \eta \dfrac{\partial E}{\partial b_j^l} & (13) \end{cases}$$

where $\eta$ is the learning rate. The selection of $\eta$ is used to control the update rate of the weights and biases.

This process above is performed iteratively until the message sequence has been recovered. The procedure of DNN-based decoding scheme is shown in Algorithm 1.

---

**Algorithm 1:** DNN-based Decoding Algorithm

---

    **Input:** input received signal $y$
    **Output:** output the recovered message sequence
1  Create input layer;
2  Create hidden layers;
3  Create output layer;
4  Initialize weights and biases;
5  **while** $i < LOOPMAX$ **do**
6     **while** $l \leq 4$ **do**
7        Calculate the input of each layer $I^l$;
8        Calculate the output of each layer $O^l$;
9        $l$++;
10    **end**
11    Calculate the loss value $E$;
12    Update weights and biases;
13    $i$++;
14  **end**

---

### 3.2. Memory Cell Analysis

For each group, the code length divided by the received signal is $n$ and the message length after decoding is $k$. In the DNN-based decoding scheme, it does not require a comparison with the Hamming codebook. As $n$ changes, it only corresponds to the weights between the input layer and the first hidden layer. If the number of neurons in the first hidden layer is $P$, the size of the weight matrix is $(P \times n)$. As $k$ changes, it only corresponds to the biases at the output layer. The size of the bias matrix is $(1 \times k)$. No extra memory cells are created during the decoding iteration due to the changes in $n$ or $k$. Thus, the total memory cells of the DNN-based decoding scheme are $(Pn + k)$.

On the other hand, in the hard decision, the received codeword is required to be compared with the codebook and the codeword that gives the minimum Hamming distance is selected. To compare the minimum Hamming distance, the size of the Hamming distance matrix is $(1 \times 2^k)$. The codebook has $2^k$ rows with $n$ lines per row. Therefore, the total memory cells of hard decision are $2^k(n + 1)$, as shown in Table 3.

**Table 3.** List of memory cells.

| Different Decoding Schemes | Size |
| --- | --- |
| DNN-based decoding | $Pn + k$ |
| Hard decision | $2^k(n + 1)$ |

Note that, as the message length $k$ of linear codes increases, the exponentiation in hard decision becomes particularly large, which will require especially large memory cells. Therefore, as $k$ increases, the number of memory cells in our proposed scheme is much lower than that of hard decision.

## 4. Simulation Results

In this section, we present some simulation results of our proposed DNN-based decoding scheme. The training platform for the DNN-based decoding scheme in simulation is established by TensorFlow [22]. The generator matrix of the (7, 4) Hamming code employed in our study is shown below.

$$\begin{bmatrix} 1 & 0 & 0 & 0 & 1 & 0 & 1 \\ 0 & 1 & 0 & 0 & 1 & 1 & 1 \\ 0 & 0 & 1 & 0 & 1 & 1 & 0 \\ 0 & 0 & 0 & 1 & 0 & 1 & 1 \end{bmatrix} \tag{14}$$

In Table 4, we present the number of neurons in the input layer, hidden layers, and output layer of the DNN model.

**Table 4.** List of the number of neurons in each layer.

| Name | The Number of Neurons |
| --- | --- |
| Input layer | 7 |
| Hidden layers | 140 |
| Output layer | 4 |

Note that the number of layers in the hidden layer is 3 and the optimization process is accomplished by the GD algorithm. Our proposed DNN-based decoding scheme is limited by the influence of various parameter settings. To identify the parameters that are most effective in reducing the bit error rate (BER), we conducted simulation analyses to observe their impact on the BER.

### 4.1. Effect of Different Learning Rates on BER Performance

In this section, the activation function of each neuron is set as sigmoid and the loss function is set as MSE. The learning rate in DNNs determines the scale of parameter adjustments during each iteration, impacting the speed and stability of the learning process. Let the learning rate $\eta$ in the GD algorithm take on the values of 0.01, 0.05, 0.08, and 0.1, respectively. In Figure 4, it is shown how the variation of learning rates affects the BER performance of the DNN-based decoding scheme. When $E_b/N_0$ is within the range of 0–3 dB, the BER curves for the four learning rates are almost identical. However, when the $E_b/N_0$ exceeds 3 dB, the four curves show significant differences. As $E_b/N_0$ increases, we can observe that the curve with $\eta = 0.08$ decreases at the fastest rate. This indicates that $\eta = 0.08$ is the most suitable learning rate for DNN-based decoding among these four curves, while keeping the other parameters constant. Moreover, the BER curve does not continuously increase with the increase in $\eta$. When $\eta = 0.10$, the curve is higher than that of $\eta = 0.08$. An excessively large $\eta$ may cause the GD algorithm to iterate repeatedly near the optimal solution, resulting in the possibility of skipping the optimal solution and encountering non-convergence regions. For a too small $\eta$, it will cause slow parameter updates, requiring more iterations to converge to the optimal solution, thereby increasing training time and computational cost. Additionally, a learning rate that is too small may result in the GD algorithm being trapped in local optima and unable to reach the global optimal solution, as there may be multiple local optima in the parameter space.

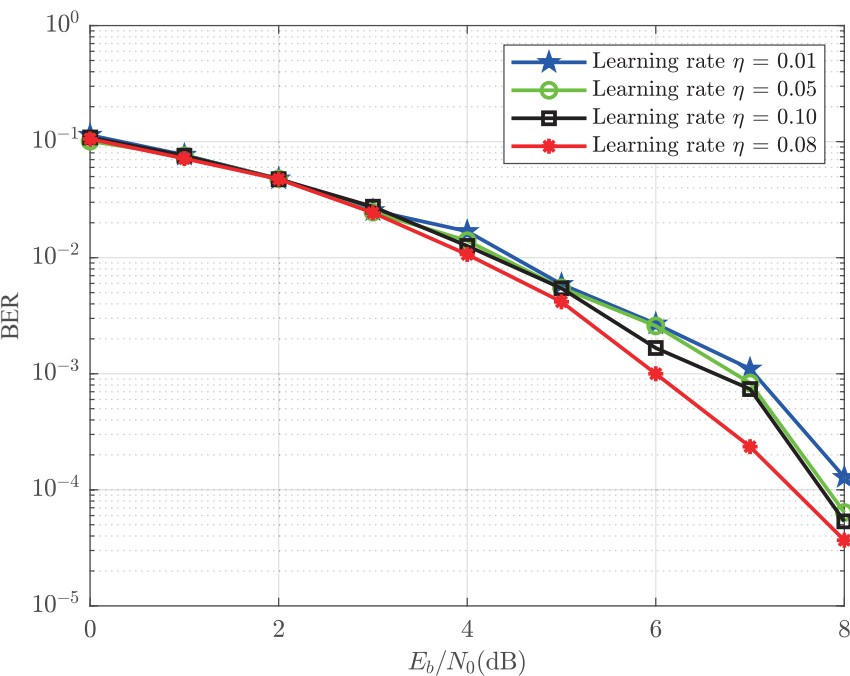

**Figure 4.** BER performance of the DNN-based decoding scheme with different learning rates.

### 4.2. Effect of Different Loss Functions on BER Performance

In this section, the activation function of each neuron is set as sigmoid and the learning rate $\eta$ in the GD algorithm is set as 0.08. The loss function guides DNNs to minimize the discrepancy between predicted output and actual target during training. Let the loss functions employed in the simulation be MAE and MSE, as mentioned in Table 2. In Figure 5, the BER performance of the DNN-based decoding scheme is compared using MAE and MSE loss functions. We can observe that the difference in the BER values between these two loss functions is not significant when $E_b/N_0$ is within 0–3.5 dB. As $E_b/N_0$ increases, the performance of the MSE and MAE loss functions in terms of BER gradually diverges. This phenomenon can be attributed to the fact that, when the signal power is significantly higher than the noise power, the MSE loss function can more sensitively

capture the slight difference between the predicted value and the true value, leading to more precise model optimization and minimized errors. On the other hand, the MAE loss function may not be sensitive enough to minor differences and not effectively optimize the model.

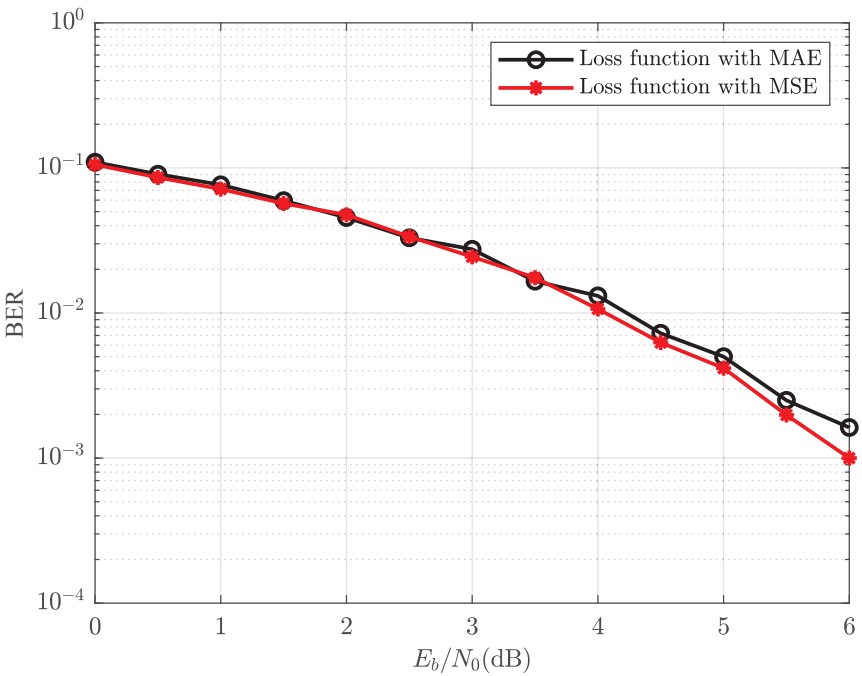

**Figure 5.** BER performance of the DNN-based decoding scheme with different loss functions.

### 4.3. Effect of Different Activation Functions on BER Performance

In this section, we employ the MSE loss function and set the learning rate to 0.08 in the GD optimization algorithm. The activation function in DNNs introduces nonlinearity to enhance their suitability for channel decoding requirements. Let the activation functions employed in the simulation be linear, tanh, sigmoid, and relu, as mentioned in Table 1. The effects of different activation functions on the BER performance of the DNN-based decoding scheme are plotted in Figure 6. The activation functions of the first and second hidden layers are set with corresponding labels in the figure, while the activation functions of the third layer in the hidden layer and the output layer are both set to the sigmoid function. The advantage of using the sigmoid function as the activation function is that it maps the neural network's output into the range of $(0, 1)$. This corresponds to the 0 and 1 in the binary sequence generated by the source and the encoded sequence. It can be observed from Figure 6 that, starting from $E_b/N_0 = 0$ dB, the BER curve corresponding to the use of the sigmoid activation function for all neurons is the lowest. The combination of relu and sigmoid functions as well as the combination of tanh and sigmoid functions show minor differences in the BER curves, while the BER curve corresponding to the combination of linear function and sigmoid function is the highest. The reason why the linear function performs poorly is because its derivative is constant, limiting the network's ability to learn complex nonlinear patterns. Nonlinear activation functions, such as sigmoid, relu, and tanh, provide stronger nonlinear transformation capabilities and improve the network's performance. Therefore, in practical applications, these nonlinear activation functions are often used to replace linear functions.

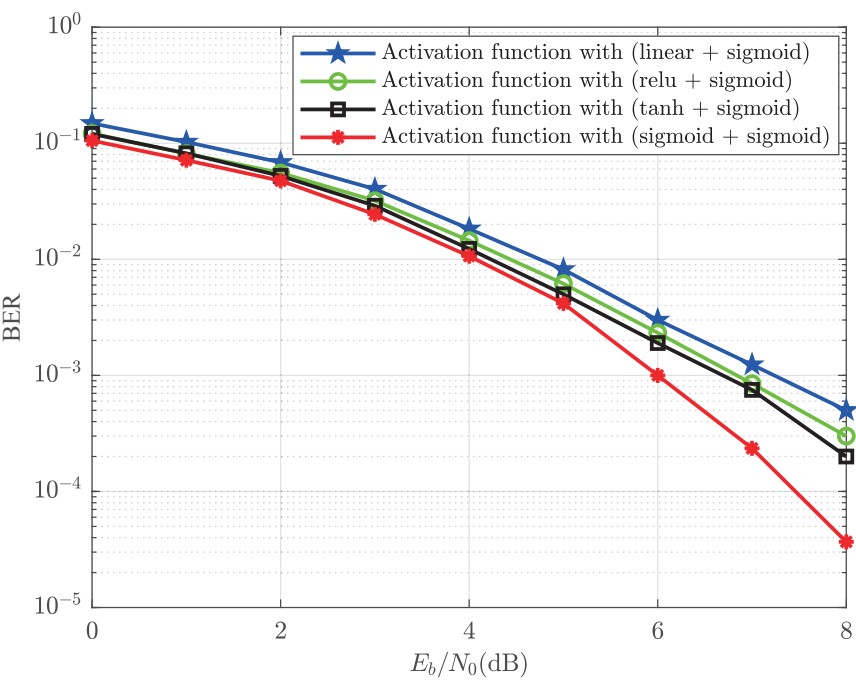

**Figure 6.** BER performance of the DNN-based decoding scheme with different activation functions.

*4.4. BER Performance Comparison of Different Decoding Schemes*

In this section, based on the simulation results of the learning rate in the GD optimization algorithm, loss function, and activation function mentioned above, the BER comparison curves between the DNN-based decoding scheme and the conventional hard-decision decoding scheme are plotted in Figure 7. For the DNN-based decoding scheme, the learning rate $\eta = 0.08$ is set to 0.08, the loss function is MSE, and the activation function for all neurons is sigmoid. The simulation results indicate that the DNN-based decoding scheme outperforms the hard decision scheme in terms of BER, demonstrating lower values over the range of $E_b/N_0$ from $-4$ dB to 8 dB. Moreover, this phenomenon verifies the feasibility of our proposed scheme.

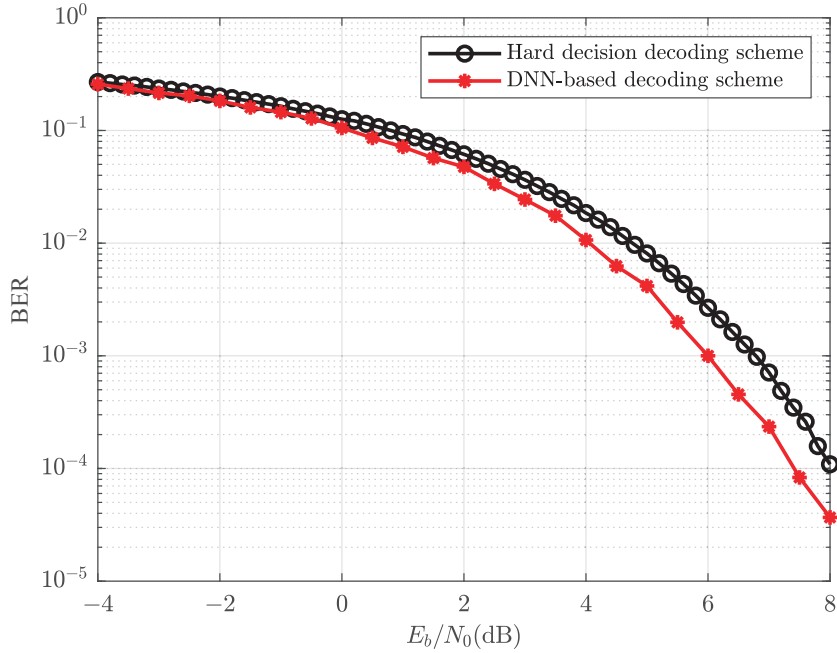

**Figure 7.** BER performance comparison of different decoding schemes.

*4.5. Memory Cell Requirements Comparison Between Our Proposed Scheme and Hard Decision*

In this section, our work can also be analyzed in terms of memory cell requirements. In practice, the required memory cells can be calculated based on Table 3. The memory cells required by hard decision and the DNN-based decoding scheme are denoted as $M_{hard}$ and $M_{DNN}$, respectively. Let us compare the memory cells of these two schemes.

When employing the (7, 4) Hamming code, the memory cell requirements for the hard decision are $M_{hard} = 2^4 \times (7 + 1) = 128$. On the other hand, the memory cell requirements for the DNN-based decoding scheme are $M_{DNN} = 140 \times 7 + 4 = 984$. Here, the number of neurons in the first hidden layer is $P = 140$. It can be obtained that the memory cell requirements for our proposed DNN-based decoding scheme are greater than those required for the traditional hard decision. However, when employing the (15, 11) Hamming code, the situation changes significantly. The memory cell requirements for these two schemes are $M_{hard} = 2^{11} \times (15 + 1) = 32{,}768$ and $M_{DNN} = 140 \times 15 + 11 = 2111$. The memory cell requirements of the hard-decision decoding scheme increase exponentially, while those of the DNN-based decoding scheme are in product form. When $k$ increases to 26, $M_{hard} = 2^{26} \times (31 + 1) = 2.1475 \times 10^9$ and $M_{DNN} = 140 \times 31 + 26 = 4366$, our proposed scheme requires about $10^5$ times fewer memory cells than hard decision, as shown in Table 5 .

**Table 5.** List of the memory cell requirements comparison for different decoding schemes.

| Different Hamming Codes | Hard Decision | DNN-Based Decoding |
| --- | --- | --- |
| n = 31, k = 26 | $2.1475 \times 10^9$ | 4366 |
| n = 15, k = 11 | 32,768 | 2111 |
| n = 7, k = 4 | 128 | 984 |

## 5. Conclusions

We proposed a DNN-based decoding scheme and demonstrated its effectiveness through a practical case study. Based on the communication transmission system model, the proposed DNN-based decoding scheme could replace the conventional channel decoding scheme. DNN models have the ability to learn through deep nonlinear transformations, which enhance the signal decoding capability and consequently achieve a lower bit error rate. Specifically, after receiving the signal from the channel, it was fed to the input layer of the DNN model at the receiver. After multiple rounds of local decoding, i.e., linear and nonlinear transformations, the predicted sequence was obtained in the output layer during this iterative process. To assess the correctness of the predicted sequence in each iteration, we introduced the MSE loss function to measure the loss value. Based on the evaluated loss value, we applied the GD algorithm to optimize the weight and bias parameters of the DNN. After numerous iterations aimed at minimizing the loss value, we recovered the near-optimal message sequence. In the simulation, we considered various factors that could impact the performance of the DNN. A detailed analysis was conducted to assess the impact of each factor, including learning rate, loss function, and activation function, on decoding performance. The optimal combination of these three factors was carefully selected for the final simulation of BER. The simulation showed that our proposed DNN-based decoding scheme was superior to the conventional hard decision decoding scheme. Moreover, this demonstrated the feasibility of our proposed scheme.

In this paper, we proposed a novel decoding scheme that integrated the DNN model with communication transmission systems. Our hope is that this decoding scheme, which is not limited to mathematical models, can inspire the application of DNNs in the field of wireless communication. An interesting problem for future research is how to jointly integrate DNNs with channel coding, channel transmission, and channel decoding. Furthermore, in real-world scenarios, the deployment of the DNN-based decoding scheme is not only constrained by learning rates, loss functions, and activation functions. Factors

such as computational resources, the availability of training data, and the requirement for hardware acceleration also play equally important roles.

**Author Contributions:** Conceptualization: Y.L., M.H.; methodology: Y.L., H.S.; software: Y.L., X.T.; formal analysis: M.H., Z.H.; investigation: Y.L., M.H., H.S.; resources: Y.L., M.H., Z.H.; data curation: Y.L., Z.H., P.P., H.W.; writing—original draft preparation: Y.L., M.H.; writing—review and editing: P.P., H.W.; visualization: X.T., P.P.; supervision: H.W.; project administration: M.H.; funding acquisition: M.H., P.P., H.W. All authors have read and agreed to the published version of the manuscript.

**Funding:** This work was supported in part by Zhejiang Provincial Natural Science Foundation of China under Grant No. LQ20F010007, No. LY22F010012, and in part by National Natural Science Foundation of China under Grant No. 62101169.

**Data Availability Statement:** Not applicable.

**Acknowledgments:** The authors would like to thank Zhejiang Provincial Natural Science Foundation of China, National Natural Science Foundation of China, Zhejiang Provincial Education Department and Hangzhou Dianzi University for support this work.

**Conflicts of Interest:** The authors declare that there is no conflict of interests regarding the publication of this paper.

## Abbreviations

The following abbreviations are used in this manuscript:

| | |
|---|---|
| DNNs | Deep neural networks |
| DL | Deep learning |
| MIMO | Multiple input multiple output |
| AWGN | Additive white Gaussian noise |
| MSE | Mean squared error |
| MAE | Mean absolute error |
| GD | Gradient descent |
| BPSK | Binary phase shift keying |
| BER | Bit error rate |

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
