# Peer review of "A Deep-Neural-Network-Based Decoding Scheme in Wireless Communication Systems†"

_electronics, doi:10.3390/electronics12132973_

Round 1

Reviewer 1 Report

The abstract is missing the following.

Problem statement and motivation

Result analysis with significant achieved results

Related work is completely missing in the manuscript. 

The latest references in the manuscript are completely missing. 

I suggest the authors to present the few more examples of the model, as taking the more data.

A character "c" in conclusion heading in small, make it capital.

The conclusion must be supported by the results.

This paper requires the minor modifications and final proof-reading. 

Reviewer 2 Report

The authors have proposed the deep neural network based decoder for wireless communications, however there are few concerns need to be addressed:

1. It is found that the authors have copied the abstract from their published work, which is available online at https://ieeexplore.ieee.org/document/9940380. So, the authors need to modify the abstract in the manuscript by avoiding the published data in it.

2. There are grammatical mistakes in the manuscript such as "conclusion" heading of section 5. The authors need to check the entire manuscript and rectify these type of errors.

3. The authors need to provide the system environment they used to train their model. 

4. The authors need to clarify about the accuracy. As they mentioned that their proposed work achieved higher accuracy. In addition to that they need to explain about the performance evaluation parameters used for their model.

The authors have proposed the deep neural network based decoder for wireless communications, however there are few concerns need to be addressed:

1. It is found that the authors have copied the abstract from their published work, which is available online at https://ieeexplore.ieee.org/document/9940380. So, the authors need to modify the abstract in the manuscript by avoiding the published data in it.

2. There are grammatical mistakes in the manuscript such as "conclusion" heading of section 5. The authors need to check the entire manuscript and rectify these type of errors.

Reviewer 3 Report

In this manuscript, the authors apply deep learning to wireless communications. They propose a reliable decoding scheme based on deep neural networks (DNNs) for communication transmission systems. Traditional channel decoding schemes are inefficiency and have limitations, particularly when faced with complex communication scenarios. The proposed DNN-based decoding scheme has ability to learn from input data to output results through end-to-end learning. It also shows potential for parallel computation, which can leverage hardware accelerators for faster training and inference speeds. The results are interesting and publishable. There are some comments for the authors to consider.

1. Could the authors provide more clear motivation and background? Please further explain why traditional decoding schemes may have limitations or drawbacks, and highlight the potential advantages of using DNN scheme accordingly.

2. While BER is a commonly used performance metric, could the authors consider including other metrics like block error rate (BLER) or frame error rate (FER) to provide a more comprehensive evaluation of the decoding scheme's performance?

3. Please further povide practical implications: Discuss the potential practical implications and challenges of implementing the DNN-based decoding scheme in real-world scenarios. Consider factors such as computational resources, training data availability, and the need for hardware acceleration to make informed decisions about the scheme's feasibility and deployment.  

4. If the authors could provide more detailed figure legends, that will facilitate readers’ understanding.

English language is almost fine. If making some minor editing of English language, that would be better.
